# Effect of Collagen Crosslinkers on Dentin Bond Strength of Adhesive Systems: A Systematic Review and Meta-Analysis

**DOI:** 10.3390/cells11152417

**Published:** 2022-08-04

**Authors:** Louis Hardan, Umer Daood, Rim Bourgi, Carlos Enrique Cuevas-Suárez, Walter Devoto, Maciej Zarow, Natalia Jakubowicz, Juan Eliezer Zamarripa-Calderón, Mateusz Radwanski, Giovana Orsini, Monika Lukomska-Szymanska

**Affiliations:** 1Department of Restorative Dentistry, School of Dentistry, Saint-Joseph University, Beirut 1107 2180, Lebanon; 2Clinical Dentistry, Restorative Division, Faculty of Dentistry, International Medical University Kuala Lumpur, 126, Jalan Jalil Perkasa 19, Bukit Jalil, Wilayah Persekutuan, Kuala Lumpur 57000, Malaysia; 3Dental Materials Laboratory, Academic Area of Dentistry, Autonomous University of Hidalgo State, Circuito Ex Hacienda La Concepción S/N, San Agustín Tlaxiaca 42160, Mexico; 4Independent Researcher, 16030 Sestri Levante, Italy; 5“NZOZ SPS Dentist” Dental Clinic and Postgraduate Course Centre, pl. Inwalidow 7/5, 30-033 Cracow, Poland; 6Department of Endodontics, Chair of Conservative Dentistry and Endodontics, Medical University of Lodz, 251 Pomorska St., 92-213 Lodz, Poland; 7Department of Clinical Sciences and Stomatology, School of Medicine, Polytechnic University of Marche, Via Tronto 10, 60126 Ancona, Italy; 8Department of General Dentistry, Medical University of Lodz, 251 Pomorska St., 92-213 Lodz, Poland

**Keywords:** aging, collagen, dentin-bonding agents, proanthocyanidins

## Abstract

This study aimed to identify the role of crosslinking agents in the resin–dentin bond strength (BS) when used as modifiers in adhesives or pretreatments to the dentin surface through a systematic review and meta-analysis. This paper was conducted according to the directions of the PRISMA 2020 statement. The research question of this review was: “Would the use of crosslinkers agents improve the BS of resin-based materials to dentin?” The literature search was conducted in the following databases: Embase, PubMed, Scielo, Scopus, and Web of Science. Manuscripts that reported the effect on the BS after the use of crosslinking agents were included. The meta-analyses were performed using Review Manager v5.4.1. The comparisons were performed by comparing the standardized mean difference between the BS values obtained using the crosslinker agent or the control group. The subgroup comparisons were performed based on the adhesive strategy used (total-etch or self-etch). The immediate and long-term data were analyzed separately. A total of 50 articles were included in the qualitative analysis, while 45 articles were considered for the quantitative analysis. The meta-analysis suggested that pretreatment with epigallocatechin-3-gallate (EGCG), carbodiimide, ethylenediaminetetraacetic acid (EDTA), glutaraldehyde, and riboflavin crosslinking agents improved the long-term BS of resin composites to dentin (*p* ≤ 0.02). On the other hand, the use of proanthocyanidins as a pretreatment improved both the immediate and long-term BS values (*p* ≤ 0.02). When incorporated within the adhesive formulation, only glutaraldehyde, riboflavin, and EGCG improved the long-term BS to dentin. It could be concluded that the application of different crosslinking agents such as carbodiimide, EDTA, glutaraldehyde, riboflavin, and EGCG improved the long-term BS of adhesive systems to dentin. This effect was observed when these crosslinkers were used as a separate step and when incorporated within the formulation of the adhesive system.

## 1. Introduction

For years, there have been many efforts to accomplish the suitable and predictable adhesion of composite resins to the dental structure, since reliable bonds offer high retention strength, less microleakage, and restoration stability [1]. Collagen fibrils aid in anchoring resin composite restorations to the dentin substrate [2]. Therefore, the long-term adhesion is a direct result of the stability and integrity of the collagen fibrils within the resin–dentin hybrid layer [3]. Dentin is acellular, collagen-rich, and avascular and contains type I collagen, non-collagenous proteins, and carbonated apatite. The collagen structure is flanked by C-terminal and globular N-terminal propeptides along with undeviating telopeptides, thereby accommodating the hydroxyapatite crystals and supporting the tissue [4,5,6].

It is necessary to distinguish that the mechanism of hybridization relies on the creation of a suitable hybrid layer to achieve the micromechanical retention of the dental restoration [7]. With that said, in restorative dentistry, the high mechanical properties of collagen are desirable. The resin–tooth interface should also have a lower rate of biodegradation to ensure the longevity of the restoration [8]. The improvement of the mechanical stability and physicomechanical properties of the collagen can be achieved by protecting them with several biomodifications using different collagen crosslinking agents [9]. The pretreatment of the dentin surface with these agents before bonding procedures may help increase the bond strength (BS) values and improve the stiffness of the resin–dentin bond [9,10,11,12,13,14,15]. The dentinal mineral phase might be depleted, causing denudation of the collagen scaffold, which can become simply confronted by the bacterially derived enzymes or endogenous proteases. This can lead to irreversible destruction of the dentinal structure. This led researchers to consider it critical to identify solutions for the long-term bond durability of adhesive systems [12].

Collagen fibers are fashioned from bundles of crosslinked microfibrils with endogenous crosslinks and covalent bonds. The covalent crosslinks produced with external crosslinking agents can be very stable over time, thereby inactivating the active sites of the dentin proteases by reducing the molecular mobility and inducing conformational changes in their structure or changing the negatively charged ionized carboxyl groups to positively charged amides. This approach aims to strengthen the collagen fibrils in the hybrid layer via intermolecular crosslinking [10,11]. Intriguingly, it is well acknowledged that the stiffness of the collagen matrix decreases from 18,000 MPa in the mineralized tissue to 1–3 MPa in the demineralized tissue. This low elasticity modulus permits further movements, such as lateral and rotational movements of the adjacent collagen peptides, taking them within the extent of the active site of the matrix metalloproteinases (MMPs), which is an expected consequence of the dentin collagen degradation [16]. Correspondingly, many natural or synthetic crosslinking agents have been claimed to offer significant advantages in developing mechanically stable collagen scaffolds [12]. It should be noted that crosslinking agents yield crosslinks in collagen fibers that reinforce the scaffold sufficiently. Consequently, unwinding its structure becomes impossible. Further, crosslinking agents can crosslink proteases, which truthfully interferes with their molecular mobility; they can disable the C-terminal telopeptides, thereby maintaining the telopeptides’ aptitude to sterically block collagenase binding to the critical peptide bond. Thus, a comparison of each agent may be helpful to select the most effective one for everyday clinical practice [17].

However, this issue is still controversial, and it is unclear in the literature which crosslinking agents could be effective for application to the dentin substrate while performing contemporary clinical bonding procedures. Suitably, systematic reviews are important tools that can provide a map of the range of available evidence. Hence, this study aimed to identify whether the BS to dentin could be affected by modifying or pretreating the dentin with crosslinking agents through a systematic review and meta-analysis. Hence, the null hypothesis to be tested was that there is no effect on the immediate and long-term dentinal BS levels when a collagen crosslinker is used for the bonding procedures.

## 2. Materials and Methods

This systematic review and meta-analysis was conducted according to the directions of the PRISMA 2020 statement [18]. The protocol was registered in the Open Science Framework with the registration number 0000-0002-2759-8984. The following PICOS strategy was used: population, dentin substrate; intervention, use of collagen crosslinkers; control, adhesive application without the use of a collagen crosslinker; outcome, BS; study design, in vitro studies. The research question of this review was: “Does the use of crosslinker agents improve the BS of resin-based materials to dentin?”

### 2.1. Literature Search

The literature search was conducted until 24 May 2021. The following databases were screened: Embase, PubMed, Scielo, Scopus, and Web of Science. The search strategy and keywords used in PubMed are listed in Table 1, as this search strategy was adapted to be used in the other screened databases. After completing the search, all articles were imported into EndNote X7 software (Thomson Reuters) to eliminate duplicates. After this, the articles were imported into the Rayyan QCRI mobile app [19].

### 2.2. Study Selection

Two reviewers (C.E.C.-S. and R.B.) independently assessed the titles and abstracts of all studies. Manuscripts for full-text review were chosen in line with the inclusion and exclusion criteria depicted in Table 2. A full text was considered if there were insufficient data to make clear decisions. A structured and logical approach to the literature search was used to identify the relevant papers reporting the effect of crosslinking agents. Reference lists of the original studies were hand-searched to identify any articles that could have been missed during the initial search, keeping the inclusion criteria in mind. Those articles that seemed to meet the inclusion criteria or had inadequate data in the title and abstract to make a clear decision were selected for a full evaluation. Any disagreement or discrepancy regarding the eligibility of the included manuscripts was resolved and decided through consensus and agreement by a third reviewer (L.H.). The reviewers also hand-searched the list of references for the included articles to identify additional articles not found in the initial search.

### 2.3. Data Extraction

Data of relevance from the studies involved were extracted using Microsoft Office Excel 2019 sheets (Microsoft Corporation, Redmond, WA, USA). These data comprised the study and year of publication, the type of tooth, the collagen crosslinker used, the mode of application of the crosslinker, the BS test used, the aging conditions for the BS test, the adhesive system used, the predominant failure mode evaluated, and the main results. If any data were missing or unclear, the corresponding author of the study was contacted via e-mail to retrieve the missing information. If the investigators did not respond within 2 weeks, the missing information was not included.

### 2.4. Quality Assessment

The methodological quality of each included manuscript was assessed independently by two authors (L.H. and R.B.). The risk of bias in each article was assessed via the description of the following parameters: specimen randomization, single-operator protocol implementation, operator-blinded, presence of a control group, standardized samples, failure mode, manufacturer’s instructions, and sample size calculation [1]. If the article included the parameter, the study received a “YES” for that specific parameter. In the case of missing data, the parameter received a “NO”. The risk of bias was classified regarding to the sum of “YES” answers received: 1 to 3 indicated a high bias, 4 to 6 medium, and 7 to 8 indicated a low risk of bias.

### 2.5. Statistical Analysis

Meta-analyses were performed using Review Manager v5.4.1 (The Cochrane Collaboration) software program. The comparisons were performed using the random-effects model, and the standardized mean difference between the BS values obtained using the crosslinker agent or the control group. Subgroup comparisons were made depending on the adhesive strategy used (total-etch or self-etch). Immediate and long-term BS data were analyzed separately. Additionally, a different analysis was performed when the crosslinking agent was used as a primer or was incorporated within the adhesive formulation. If the study compared several experimental groups against the same control group, data for the experimental groups (mean, standard deviation (SD), and sample size) were combined for the meta-analysis. The heterogeneity among studies was assessed using the Cochran Q test and the I2 test.

## 3. Results

A total of 3071 papers were retrieved from all databases. After eliminating the duplicates, 2147 documents were examined by reading the title and abstract. After this, 2037 articles were excluded because they did not meet the inclusion criteria, leaving a total of 110 articles to be evaluated by full-text reading. Of these articles, 60 were not considered in the qualitative analysis: 1 article was in a language other than English [20], 2 studies were clinical trials [21,22], 1 study evaluated the BS only to carious dentin [23], 3 articles used the crosslinker agent within the composition of the phosphoric acid [24,25,26], 13 studies evaluated experimental adhesives [12,14,16,27,28,29,30,31,32,33,34,35,36], for 5 articles we could not retrieve the full text [37,38,39,40,41], 25 articles did not evaluate any BS test [42,43,44,45,46,47,48,49,50,51,52,53,54,55,56,57,58,59,60,61,62,63,64,65,66], 8 articles did not use a crosslinker agent [25,67,68,69,70,71,72,73], 1 article was a review [74], and 1 article tested the BS in bovine teeth as a substrate [75]. A total of 50 articles were included in the qualitative analysis. Of these, 5 articles were not considered for the meta-analysis because two of them did not express the BS data as means and SDs [76,77], and in 3 articles the number of specimens used could not be retrieved [78,79,80], totalling 45 articles considered for the quantitative analysis [17,81,82,83,84,85,86,87,88,89,90,91,92,93,94,95,96,97,98,99,100,101,102,103,104,105,106,107,108,109,110,111,112,113,114,115,116,117,118,119,120,121,122,123,124,125]. A flowchart designating the study selection process according to the PRISMA statement is presented in Figure 1.

The characteristics of the studies included in the qualitative analysis are shown in Table 3.

This review identified several substances proposed as crosslinking agents, including carbodiimide, epigallocatechin-3-gallate (EGCG), glutaraldehyde, riboflavin, proanthocyanidins, ethylenediaminetetraacetic acid (EDTA), benzalkonium chloride, zinc, chitosan, minocycline, hesperidin, myricetin, sodium ascorbate, and galardin. The crosslinking agents were used as primers or incorporated into the adhesive formulation.

The meta-analysis was conducted considering 45 articles. Separate analyses were performed for the crosslinking agents used as pretreatments or incorporated within the adhesive formulation. Additionally, separate analyses were performed for the immediate and long-term bond strengths. Separate subgroups were analyzed for each condition considering the type of adhesive used (self-etch or total-etch).

Figure 2, Figure 3, Figure 4, Figure 5, Figure 6, Figure 7 and Figure 8 show the analyses of the crosslinkers agents when applied as a separate step as a primer. 

When chitosan was used prior to the application of an adhesive system, neither the immediate nor long-term BS values were improved (Figure 2; *p* = 0.3, and *p* = 0.97).

Figure 3 shows the effects of the application of the EGCG crosslinker agent. The meta-analysis favoured the use of this crosslinking agent only for the long-term BS (*p* = 0.03) while an improvement in the immediate BS was not observed (*p* = 0.79). This behaviour was found for the carbodiimide (Figure 4; long-term *p* = 0.002, immediate, *p* = 0.32), EDTA (Figure 5; long-term *p* = 0.02, immediate, *p* = 0.48), glutaraldehyde (Figure 6; immediate, *p* = 0.09; long-term, *p* < 0.00001), and riboflavin (Figure 7; immediate, *p* = 0.31; long-term, *p* < 0.00001).

Figure 8 shows a forest plot of the comparison of the proanthocyanidins used as a crosslinking agent. According to the analysis, both the immediate and long-term BS values are improved when this crosslinking agent is used (*p* ≤ 0.02).

Figure 9, Figure 10, Figure 11 and Figure 12 show the analyses of the crosslinkers agents when incorporated within the formulation of the adhesive system. According to the analyses, the incorporation of proanthocyanidins did not improve the immediate BS (Figure 9; *p* = 0.21).

On the other hand, the incorporation of glutaraldehyde (Figure 10, *p* = 0.009), riboflavin (Figure 11, *p* < 0.00001), and EGCG (Figure 12, *p* < 0.00001) into an adhesive system improved the long-term BS to dentin.

Conferring to the parameters contemplated in the risk of bias assessment, the majority of manuscripts were categorized with a medium risk of bias (Table 4). Numerous studies failed to report the single-operator, operator-blinded, and sample size calculation parameters.

## 4. Discussion

A systematic review and meta-analysis were performed regarding the immediate and long-term BS levels of adhesive systems to dentin depending on the crosslinking agents used as pretreatments or incorporated within the adhesive formulation. Some of the crosslinkers increased the BS, while others lacked any influence. Hence, the hypothesis examined in this review was partially accepted.

Demineralized collagen fibrils are susceptible to hydrolytic or enzymatic degradation over time, leaving voids or demineralized nanochannels within the resin dentin hybrid layer [11]. With time, the degree of degradation increases and is mostly concerted at the bottom part of the hybrid layer, where the collagen appears to be less penetrated by resin monomers [126] and displays elevated enzymatic activity [127]. Collagen fibrils are deemed essential for helping in anchoring the composite resin to the dentinal surface. Consequently, they might be recognized as an imperative feature for the durability of dentin adhesion [2,128]. In an attempt to improve the resin–dentin interface, numerous researchers have focused on lessening the enzymatic biodegradation through the use of MMP inhibitors [129,130,131] separately from other recommended clinically relevant approaches [132]. Biomodifications and increases in the chemical and mechanical properties of collagen are deemed essential to improve the stiffness of collagen fibers [111]. This could be possible with the use of collagen crosslinking agents.

Appropriately, several substances have been argued to offer noteworthy benefits in protecting collagen fibers, and they were proposed as crosslinking agents, including carbodiimide, EGCG, glutaraldehyde, riboflavin, proanthocyanidin, EDTA, benzalkonium chloride, zinc, chitosan, minocycline, hesperidin, myricetin, sodium ascorbate, and galardin. The crosslinking agents were used as primers or incorporated into the adhesive formulation.

The analyses of the crosslinkers agents when applied (as a separate step) as primers showed that when chitosan was used prior to the application of an adhesive system, neither the immediate nor long-term BS values were improved (*p* = 0.77, and *p* = 0.97).

One should bear in mind that chitosan is known for being biocompatible and antibacterial, and has a broad spectrum of dental applications [49,133]. It possesses an affinity to amino acids and can generate microfibrillar arrangements in the collagen structure [134]. This combination provides benefits for tissue engineering purposes [135], as type I collagen has an aptitude to form ionic complexes with chitosan, increasing the collagen strength [136]. Previous studies [137,138] have reported that the incorporation of chitosan into dentin adhesives seemingly did not have any influence on the BS of the adhesive to the dentin structure. This could be similar to the results of this study, which showed that using chitosan as a separate primer solution could not enhance the BS. This might be explained by the fact that chitosan may increase the hydrophilicity of some adhesive systems, thereby permitting the pathway of nanometer-sized defects within the hybrid layer and making it more porous. In addition, chitosan used alone was considered a weak crosslinker, since it was demonstrated to lessen the activity of collagenolytic enzymes when applied to dentin without adhesive. Further, a detrimental interaction between the chitosan and adhesive could be the reason for this BS reduction [107].

The effect of the application of the EGCG favoured only the long-term BS (*p* = 0.03), while an improvement in the immediate BS was not observed (*p* = 0.79). This behaviour was observed for the carbodiimide (long-term *p* = 0.002, immediate, *p* = 0.32), EDTA (long-term *p* = 0.02, immediate, *p* = 0.48), glutaraldehyde (immediate, *p* = 0.09; long-term, *p* < 0.00001), and riboflavin (immediate, *p* = 0.31; long-term, *p* < 0.00001).

The use of an EGCG crosslinker had been advocated to impair the immediate BS. EGCG, a major green tea polyphenol, provides numerous functions, such as anti-inflammatory, antimicrobial, anticollagenolytic, antioxidant, and anticancerogenic effects [139,140]. This crosslinker was able to make the collagen chain stable [141,142]. In addition, it can reduce the biodegradation of collagen and increase the number of collagen crosslinks through the interaction of hydrogen molecules of galloyl groups [141]. It was demonstrated that EGCG applied for 60 s was efficient in maintaining the resin–dentin bond durability. Although this application period is clinically possible, it includes adding a step to the adhesive technique, thereby increasing the clinical time of the process, which contradicts the trend of simplifying materials [95]. Previous research revealed that this crosslinker enhanced the BS of some adhesive systems [143]. However, EGCG at higher concentrations could interrupt the adhesive polymerization, thereby affecting the BS [144]. Even at a concentration of 0.1%, this crosslinker did not show promising results [95]. The EGCG is probably entrapped within the linear chains after curing, thereby interfering with the monomer conversion and BS. These data corroborate the findings of this study.

Likewise, carbodiimides are a substitute agent for crosslinking to glutaraldehyde that do not contain toxic components and show greater biocompatibility because of their urea derivative, which can be easily rinsed from collagen without leaving any residual chemicals. They enhance the mechanical properties of collagen, making it more difficult to unwind and strengthening the hybrid layer [55]. A higher carbodiimide concentration can inactivate the dentin proteases, especially MMP-9 with a one minute application period after the use of acid-etched dentin [145]. They crosslink proteins (collagen) via the activation of the carboxylic acid groups of glutamic and aspartic acid residues in peptides by donating O-acylisourea groups within a treatment time of 1 h, which is clinically intolerable [146]. Perdigão et al. reported that 1 to 4 h are needed for an effective carbodiimide application for a stronger hybrid layer. Hence, their effect is concentration-dependent [145]. Carbodiimides enhanced the resistance to fatigue cracking after 6 months of water storage in dentin-bonded specimens according to Zhang et al. in 2016 [82]. This could correlate with the result of this study.

Equally, EDTA with 4 carboxylic groups showed promising results after aging. This agent is used for irrigation during the mechanical instrumentation of the root canal system and is known to decalcify smear-layer-covered dentin. EDTA might inhibit MMP-2 and MMP-9 activity when applied for 1 to 5 min because it is an effective Zn^2+^ and Ca^2+^ chelator. This helps in improving the durability of the resin–dentin bond [147]. This conclusion seems to support the results of this meta-analysis. It also increases the BS to dentin when compared to phosphoric-acid-treated dentin [148], but it is important to mention that in 2009, Sauro et al. suggested that EDTA can be removed easily by simple water rinsing; hence, this may result in no residual EDTA remaining, consequently leading to an incapacity to inhibit the activity of MMPs [149]. It has been suggested that the efficacy of EDTA relies on numerous features, such as the hardness of the dentin structure; the duration of application depth of the material; the pH, the form (gel or liquid), and the concentration of the material [149]. Additionally, according to the meta-analysis, dentin pretreatment with EDTA improved the long-term BS. This result may have been due to an improved resin infiltration into the collagen matrix [94]. It has been previously demonstrated that EDTA improves the removal of the smear layer, and as a consequence a better interaction of the functional monomers with the dentin may occur [94,150].

With regards to glutaraldehyde, it acts as a true crosslinking agent, containing five carbon aliphatic molecules. It contains two aldehyde groups at each end of the chain able to react with the amino groups of the collagen fibrils, reinforcing the hybrid layer and making the whole adhesive interface stronger via the chemical covalent bonds formed (Schiff base crosslinks) [65]. In addition, when 5% glutaraldehyde is applied for 1 min after acid-etched dentin according to Frassetto et al. 2016 [2], this consequently leads to an increase in the dentin collagen properties. Formerly, a commercial glutaraldehyde-based desensitizer, Gluma (Heraeus Kulzer GmgH, Germany), was tested and shown to contain 5.0% glutaraldehyde, 35% hydroxy ethyl methacrylate (HEMA), and 60% water. The dentin-treated interface with Gluma revealed bond stability and positive effects; however, despite the influence of HEMA on the stability of resin–dentin bonds, previous studies confirmed that the 25% glutaraldehyde-treated dentin increased the elastic modulus and improved the resistance of collagen to degradation [45,59]. Glutaraldehyde was also successful in caries-affected dentin according to Macedo et al. [151], and promoted the remineralization of the dentin via the reaction of the carbonyl group in the glutaraldehyde and hydroxyapatite in the dentin according to Chen et al. in 2016 [152]. Despite its efficacy, glutaraldehyde is strongly toxic, thereby limiting its clinical application [153]. The shortcoming of this agent is that it causing quick surface crosslinking of the tissue, producing a barrier that hinders its supplementary dispersion into the tissue bulk, risking the fixation of the tissue since the depth of the tissue increases [154]. This was elucidated by the fact that this agent was not able to totally inhibit the activity of collagenolytic enzymes in the deeper area of the hybrid layer, meaning an improvement in the immediate BS could not be observed in some adhesive systems [17]. This conclusion seems to support the results of this meta-analysis.

Recently, physical methods, specifically ultraviolet radiation, have been tested in dentistry and ophthalmology. Riboflavin, known as vitamin B2, is a crosslinking agent that produces free radicals via photooxidation (UVA), which improves the rigidity and mechanical stability of the collagen matrix, as well as the penetration of the adhesive resin (collagen became more resistant to enzymatic degradation) [2]. Riboflavin plays an important role in adhesive dentistry due to its biocompatibility, easy application, and activation by blue light. Note that riboflavin possesses optimum absorption peaks for UVA at 270, 366, and 445 nm wavelengths. The high energy of UVA breaks down the weak intrinsic crosslinks in the collagen and generates free oxygen radicals. The latter creates a new stronger covalent bond between the collagen fibers and leads to successful results. Indeed, studies have shown that 0.1% riboflavin-treated dentin is sufficient to obtain good adhesion in bonding protocols [16,65]. Notwithstanding an immediately restricted influence in terms of the collagenolytic enzyme inhibition, the binding effect of this crosslinker can be maintained for a long time [17], as suggested also by this systematic review and meta-analysis.

According to the analysis, both the immediate and long-term BS levels were improved when the proanthocyanidins crosslinker was used (immediate, *p* = 0.02; long-term, *p* = 0.001). Proanthocyanidins are natural crosslinkers derived from polyphenols (polyphenolic compounds) that are known as tannins and could improve the mechanical properties of dentin, thereby enhancing the quality of the hybrid layer. They are prevalent in elm trees, vegetables, fruits, nuts, flowers, pine bark, and grape seeds. They are known as crosslinkers and antioxidant agents with very low toxicity [2]. In 2017, Münchow et al. suggested that the incorporation of proanthocyanidins in an experimental adhesive did not show any negative effects on the immediate resistance of the resin–dentin bond when its concentration was ≤2% [11]. However, Perdigão et al. reported that 2% proanthocyanidins had no adverse effects on the dentin BS, but 3% decreased the BS [145]. In 2016, Frassetto et al. indicated that the application of 3.75 mass% proanthocyanidins for 5, 15, or 30 s in 10-μm-deep demineralized dentin offered adequate crosslinking to protect the dentin matrix from the catalytic activity of MMP-1 and MMP-9 [2]. Another study pointed out that the interaction between collagen and proanthocyanidins is not time-dependent, even within a short time exposure of 10 s [53]. Suitably, higher concentrations of proanthocyanidins can form a denser collagen matrix, which can hinder the water leaching and decrease the vapor permeability of the proanthocyanidin–collagen film [155]. Hence, proanthocyanidins can displace the water between collagen microfibrils by creating in this way a new hydrogen bond between the fibrils, aggregating them and protecting the collagen triple helix [156]. Additionally, proanthocyanidins are extraordinarily effective in stabilizing demineralized dentinal collagen against enzymatic activity in a clinically relevant context, presumably due to the non-covalent nature and covalent, electrostatic, and hydrophobic interactions with proline-rich proteins such as collagen molecules [145]. In 2018, Gré et al. confirmed that strong bonds can be formed between the amide carbonyl group of collagens and the phenolic hydroxyl group of proanthocyanidins, leading to the formation of the proline–proanthocyanidin complex [65]. Next, in 2021, a report suggested that the biomodification of deep dentin with proanthocyanidins showed the highest shear BS, followed by riboflavin and the control group (no collagen crosslinker modification) [157]. Proanthocyanidins can be incorporated into the etching solution and experimental primer to enhance the stability of bonding, while proanthocyanidins incorporated into an experimental adhesive showed no negative effect but stabilized the interfacial resin–dentin bonds, confirming the outcome obtained in this analysis. Other studies showed a drawback of proanthocyanidins in terms of discoloration (a brownish color) of the substrate treated [65,157].

The analyses of the crosslinker agents when incorporated within the formulation of the adhesive system revealed that the incorporation of proanthocyanidins did not improve either the immediate or long-term BS (*p* = 0.21). As described before, proanthocyanidins are considered well-known crosslinkers that can enhance both the immediate and aged BS levels of pretreated dentin [105]; nevertheless, their incorporation into the self-etch primer seemed to be unsuccessful to display such an influence in this present review. As elucidated in a previous manuscript, the large molecular size of the proanthocyanidins may have impeded the etching effect of the self-etch primer, hindering the ideal hybridization between the resin monomers and dentinal structure [103]. Another study indicated a comparable effect of this crosslinker when incorporated into an adhesive system on the dentin BS [36].

On the other hand, the incorporation of glutaraldehyde (*p* = 0.009), riboflavin (*p* < 0.00001), and EGCG (*p* < 0.00001) into an adhesive system improved the long-term BS to dentin. Concerning glutaraldehyde, the mechanism appears to rely on covalent chemical bond formation between the aldehyde group of glutaraldehyde and the amino groups of hydroxylysine and lysine in collagen fibrils [158,159], which increases the ability of the collagen to resist enzymatic degradation [160]. Additionality, this agent has been able to infiltrate more deeply into the etched dentinal matrix than might the other agents, possibly enhancing the collagen crosslinking at the bottom part of the hybrid layer and diminishing the bond degradation over time, indicating that this technique has been proven as a substitute to increase the long-term BS of adhesive systems to dentin. Similar findings were observed for riboflavin incorporated into adhesive systems. This could be clarified by the fact that no competition was found between the adhesive and riboflavin in absorbing light. Furthermore, this could be explained by the crosslinking effect of riboflavin and the dual effect of not just activating the adhesive monomers, but also the riboflavin. Universal adhesives seem to benefit from the long-term 0.1% riboflavin modification, as previously reported [124]. Additionally, Daood et al. [16] presented that the 3% riboflavin incorporated into the adhesive system seems to increase the immediate BS and preserve the durability of the resin–dentinal bond, without negatively influencing the degree of conversion. Moreover, this study revealed that EGCG enhanced the bonding stability of the adhesive systems after a long aging period. A preceding manuscript also established that 200 μg/mL of an EGCG-modified adhesive system might maintain the stability of the interface between the resin monomers and dentin, and perhaps this was observed after six months of distilled water aging [144]. Further, another study evidenced that the BS of an EGCG-modified adhesive system did not diminish within six months of water storage, while a dentin pretreatment with EGCG combined with a water solution or ethanol solution may enhance the bonding performance [87]. It was recently demonstrated that the EDC crosslinking effect stayed within the hybrid layer for 5 years in terms of the BS, collagen structure maintenance, and dentinal enzyme inhibition [161]. The researchers have recognized the impact of EGCG on improving the bonding stability to the matrix-bound MMPs’ inhibition effect [84]. This could be in agreement with the findings of this study and could explain the reason behind the bond stability after aging.

The present analysis noted the effect of modifying or pretreating dentin with crosslinking agents on the bonding ability of the adhesive systems. Novel dental material types might, however, be fashioned using nanotechnologies and other new methods within the fields of biomaterials and material sciences. Such progress may include antimicrobial properties, MMP and cathepsin inhibition, collagen-strengthening properties, and dental hard tissue regeneration. In conclusion, while there are still some unsettled difficulties regarding the durability of the adhesive interface, it is truly remarkable to see how far adhesive systems have come in the past 50–60 years. Methods that are capable of creating stable resin–dentin bonds and resisting the collagenolytic hydrolysis will probably be available in the next few years, improving the quality of the dental therapies. The results of this review should be taken with caution, since some future techniques should be tested, including nanoleakage, wetting ability, scanning electron microscopy, and micro-raman spectroscopy techniques. Additionally, most of the manuscripts included were categorized as having a high or medium risk of bias, and consequently superior experimental designs should be used to acquire a higher degree of evidence. Several crosslinking agents were used to show the importance of collagen strengthening; however, many other crosslinkers were not studied and evaluated in this study. In addition, there is an opportunity for slight changes in the chemistry of these materials, causing significantly different reactions and binding with the amino acid groups of collagen fibrils. Further, one should bear in mind that the pretreatment methods might be uncontrolled, and it could be problematic to define the crosslinking effect. Supplementary studies should be carried out to investigate the intrinsic protective effects of crosslinking agents on collagen, besides their mechanism of inhibition on MMP enzymes. A previous systematic review exhibited that there is no clinical effectiveness to defending the use of collagen crosslinking agents in combination with the adhesive technique in restorative procedures [162]. This did not match the results of this systematic review; thus, randomized controlled clinical trials should be considered in future investigations. Moreover, commercial products based on crosslinkers are somewhat absent in the dental market, meaning this could be a difficulty for a dentist in their daily clinical practice. The future research directions should be focused on inventing new adhesive systems containing crosslinkers as commercial products.

The aim of introducing an additional crosslinker is to stop collagen molecules from sliding past each other under stress inside the dentinal substrate [13], which will decrease the extensibility and enhance the biomechanical strength of the collagen fibers [15]. Collagen crosslinkers were used as a substitute dentin pretreatment to improve the durability of the dentinal bonding and to strengthen the dentinal collagen fibrils [111]. This conclusion seems to match the finding of this paper, since the influence of the crosslinkers appears to develop with time to favour collagen in terms of preserving an expanded position, whereby monomers and solvents can be received inside the adhesive formulation [14].

## 5. Conclusions

Within the limitations of this systematic review, it can be concluded that the application of different crosslinking agents such as carbodiimide, EDTA, glutaraldehyde, riboflavin, and EGCG improved the long-term BS of adhesive systems to dentin. This effect was observed when these crosslinkers were used as a separate step and when incorporated within the formulation of the adhesive system.

## Figures and Tables

**Figure 1 cells-11-02417-f001:**
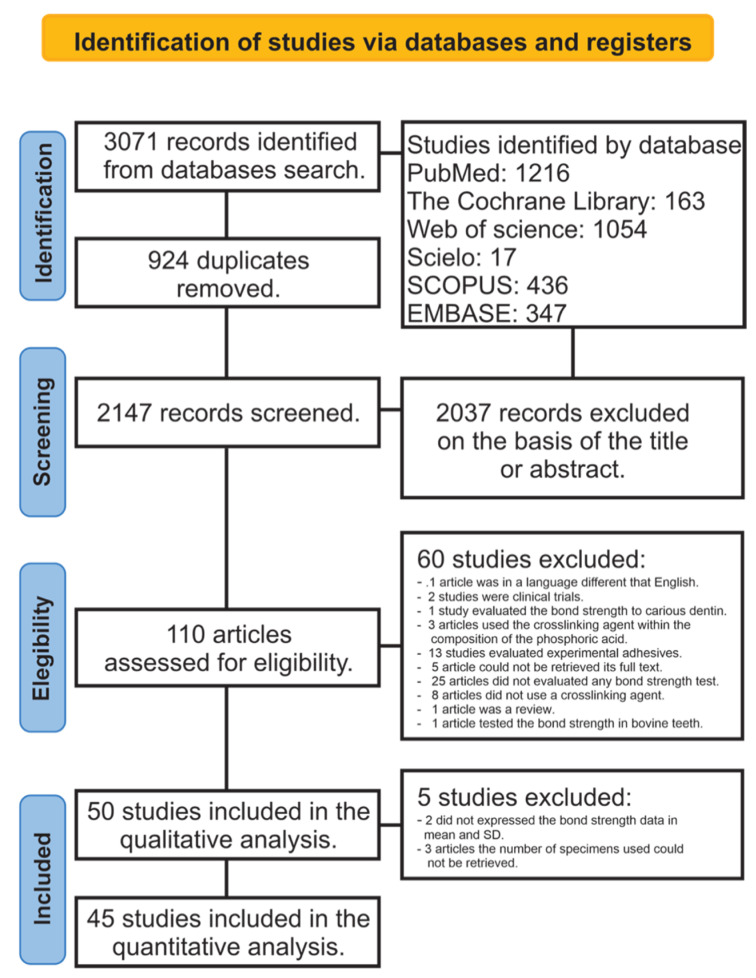
Flowchart according to PRISMA guidelines.

**Figure 2 cells-11-02417-f002:**
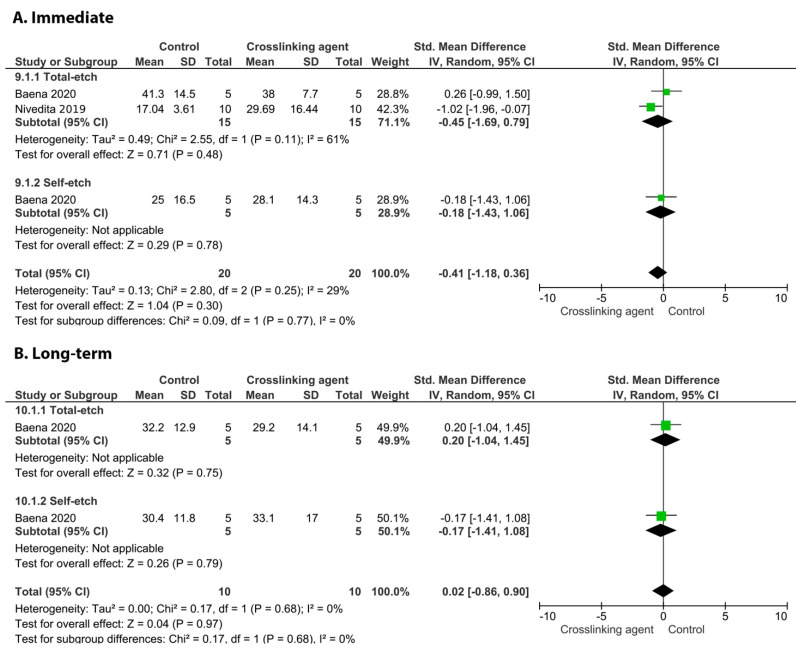
Forest plot of the immediate (**A**) and long-term (**B**) bond strength comparison between the chitosan crosslinking agent and the control according to the adhesive used [107,123].

**Figure 3 cells-11-02417-f003:**
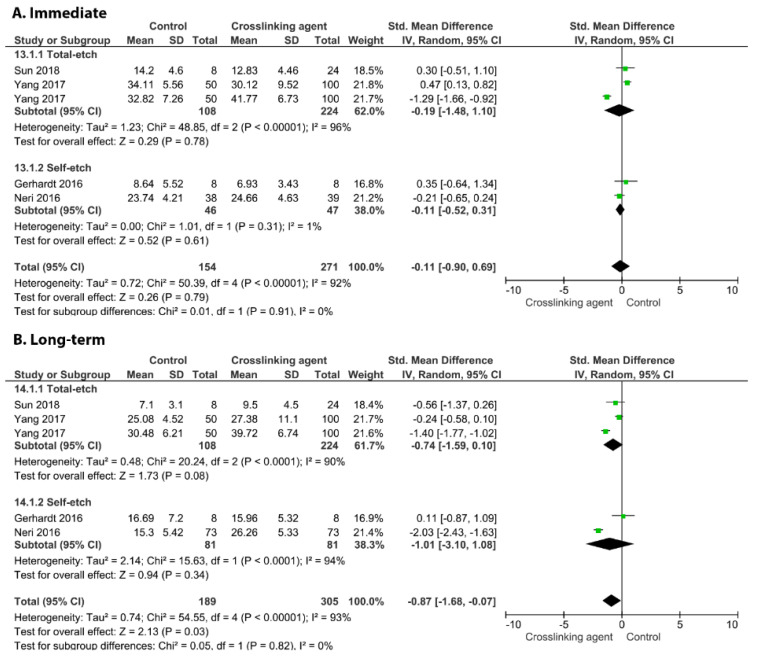
Forest plot of the immediate (**A**) and long-term (**B**) bond strength comparison between the EGCG crosslinking agent and the control according to the adhesive used [67,86,95,119].

**Figure 4 cells-11-02417-f004:**
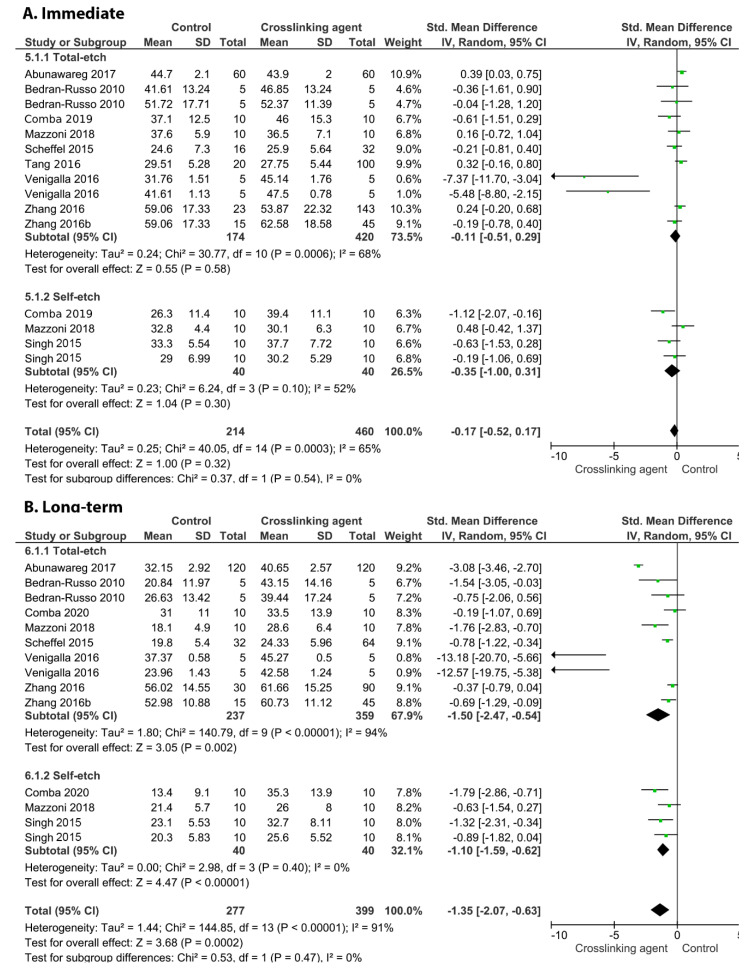
Forest plot of the immediate (**A**) and long-term (**B**) bond strength comparison between the carbodiimide crosslinking agent and the control according to the adhesive used [60,82,83,85,90,97,106,109,115].

**Figure 5 cells-11-02417-f005:**
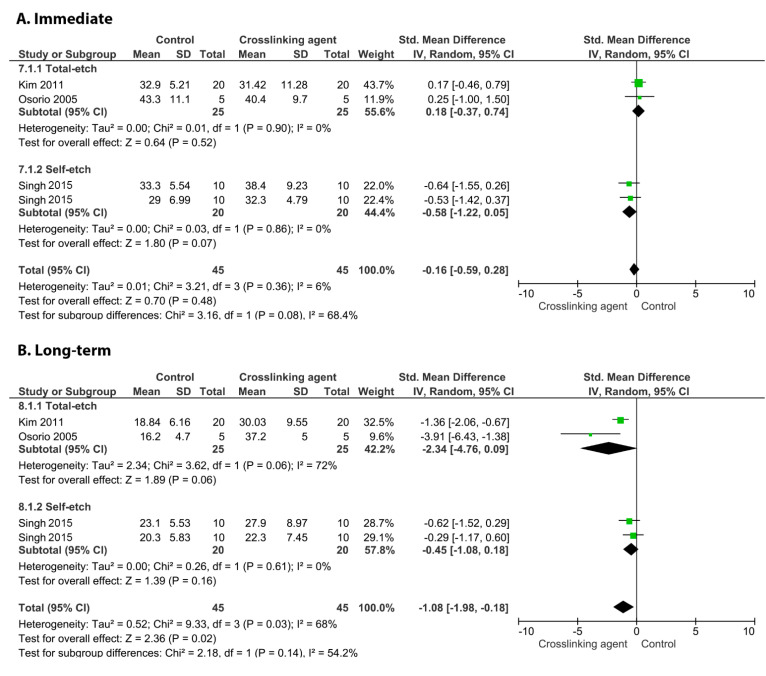
Forest plot of the immediate (**A**) and long-term (**B**) bond strength comparison between the EDTA crosslinking agent and the control according to the adhesive used [85,94,111].

**Figure 6 cells-11-02417-f006:**
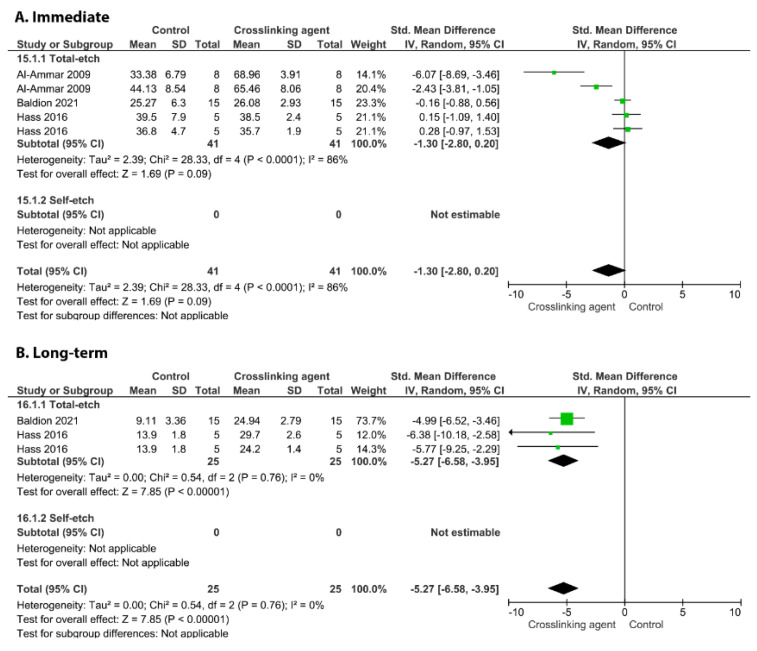
Forest plot of the immediate (**A**) and long-term (**B**) bond strength comparison between the glutaraldehyde crosslinking agent and the control according to the adhesive used [17,105,108].

**Figure 7 cells-11-02417-f007:**
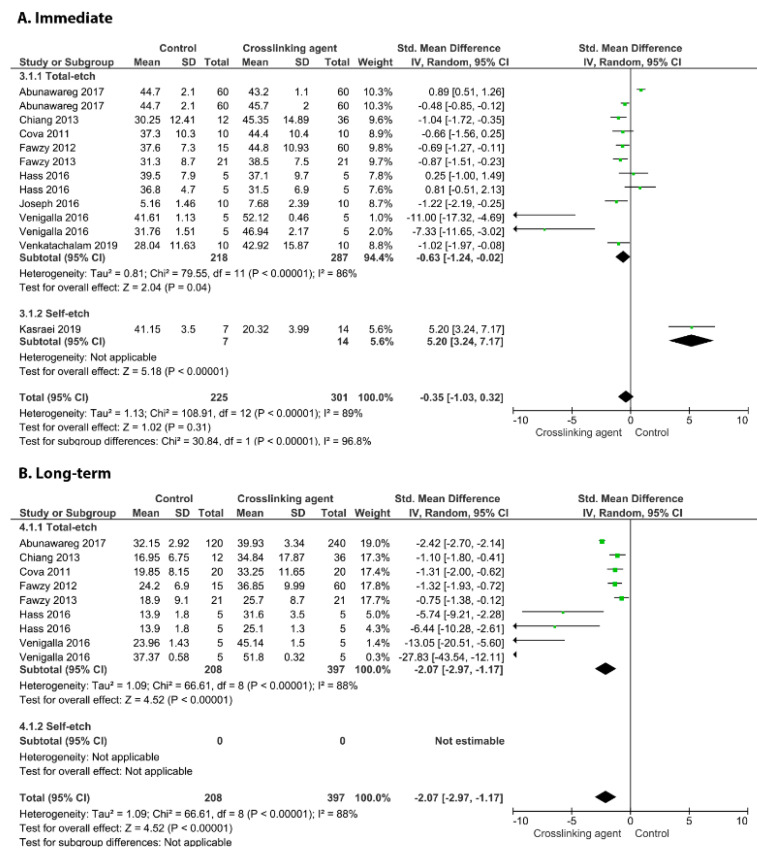
Forest plot of the immediate (**A**) and long-term (**B**) bond strength comparison between the riboflavin crosslinking agent and the control according to the adhesive used [17,83,102,106,114,116,117,118].

**Figure 8 cells-11-02417-f008:**
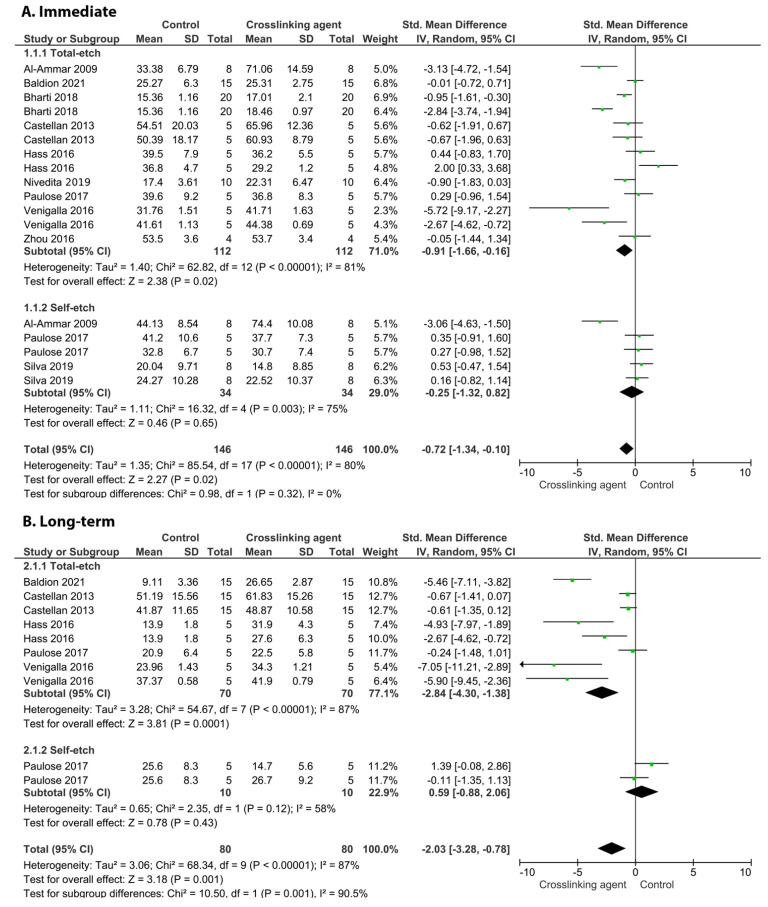
Forest plot of the immediate (**A**) and long-term (**B**) bond strength comparison between the proanthocyanidins crosslinking agent and the control according to the adhesive used [17,81,83,89,92,105,108,110,112,123].

**Figure 9 cells-11-02417-f009:**
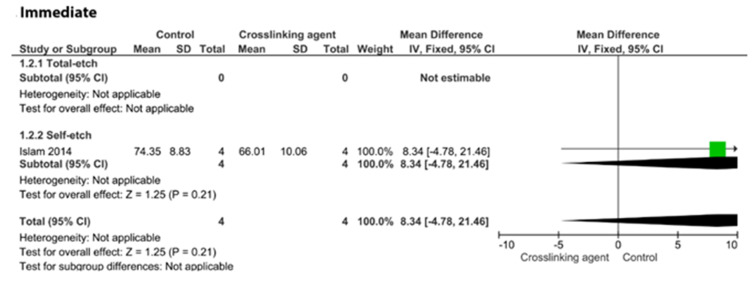
Forest plot of the immediate bond strength comparison between the proanthocyanidins crosslinking agent and the control according to the adhesive used [103].

**Figure 10 cells-11-02417-f010:**
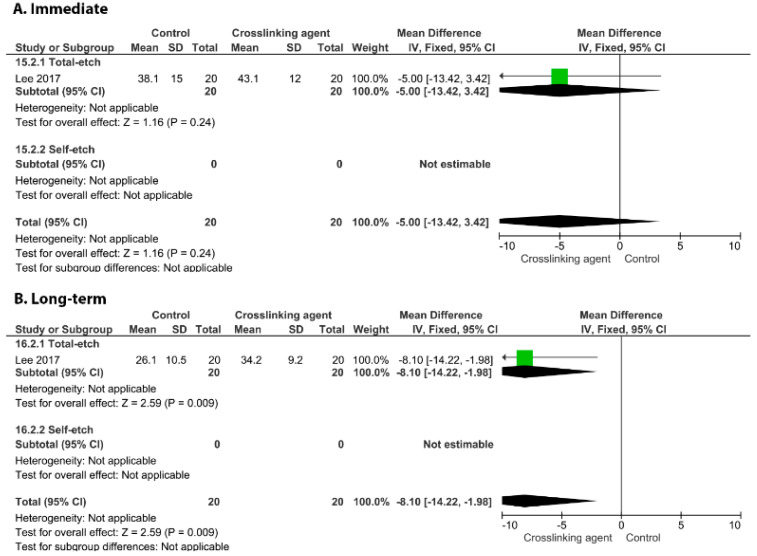
Forest plot of the immediate (**A**) and long-term (**B**) bond strength comparison between the glutaraldehyde crosslinking agent and the control according to the adhesive used [99].

**Figure 11 cells-11-02417-f011:**
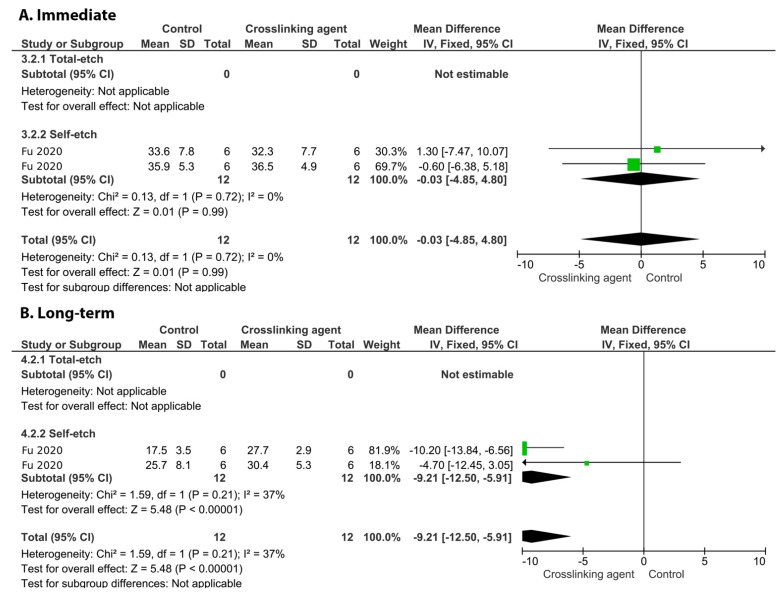
Forest plot of the immediate (**A**) and long-term (**B**) bond strength comparison between the riboflavin crosslinking agent and the control according to the adhesive used [124].

**Figure 12 cells-11-02417-f012:**
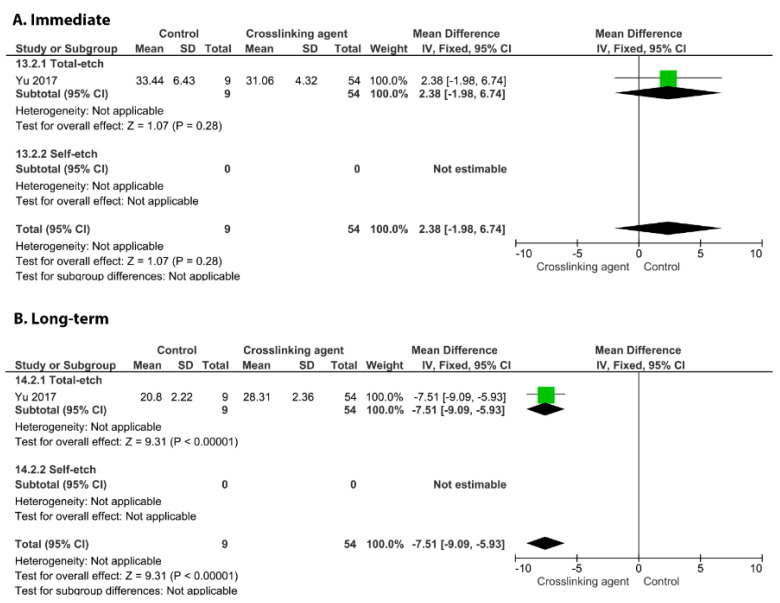
Forest plot of the immediate (**A**) and long-term (**B**) bond strength comparison between the EGCG crosslinking agent and the control according to the adhesive used [84].

**Table 1 cells-11-02417-t001:** Search strategy used in PubMed.

Search Strategy	
**# 1**	Bond OR Bonding OR Dental bonding OR Bonding efficacy OR bond strength OR Bonding performance OR bonding effectiveness OR Bond performance OR adhesive properties OR microtensile strength OR Micro-tensile strength OR bonding properties OR Microtensile bond strength OR shear bond strength OR micro shear bond strength OR performance OR Dental Bonding* OR Dentin-Bonding Agents OR Dental Bonding OR Dentin-Bonding Agents* OR Dental Bonding*
**# 2**	Dentine OR Dentin OR Dentin*
**# 3**	Cross-linking OR Cross-linkers OR Crosslinking OR Cross-links OR Cross-Linking OR cross-linking agents OR Dentin collagen OR Cross-linking agent OR Collagen Matrix OR Crosslink OR Collagen degradation OR Crosslinking and biomimicry OR Cross linker
**# 4**	#1 AND #2 AND #3

**Table 2 cells-11-02417-t002:** Inclusion and exclusion criteria.

Inclusion Criteria	Exclusion Criteria
Evaluated the bond strength to permanent human dentin of resin-based materials. Reported the effect on the bond strength of the use of a crosslinker agent prior to the application of adhesive system. Used an adhesive system modified by the incorporation of a crosslinker agent within its composition. Included a control group where a crosslinker agent was not used. Reported the bond strength in MPa.	Studies evaluating experimental adhesives. Review papers. Case reports. Commentaries. Interviews. Updates.

**Table 3 cells-11-02417-t003:** Characteristics of the studies included in the review.

Study and Year	Type of Tooth	Collagen Crosslinkers	Mode of Application	Bond Strength Test Used	Adhesive Used	Storing Conditions	Predominant Failure Mode	Main Results
**Munksgaard, 2002** **[96]**	Human teeth	HEMA and GA	Incorporated into the adhesive system	TBS	Concise Enamel Bond^®^) (3M Company, St. Paul, USA)	Distilled water at 37 °C for 24 h	NO	Bond strength was highly dependent on the HEMA and GA concentration used.
**Osorio, 2005** **[94]**	Human molars	EDTA	Pretreatment	µTBS	Adper Scotchbond 1 (3M ESPE, St. Paul, MN, USA), and Clearfil SE Bond (Kuraray Co. Ltd., Osaka, Japan)	Distilled water at 37 °C for 24 h	Mixed	Collagen network is better-preserved after EDTA application.
**Al-Ammar, 2009** **[105]**	Human molars	GA, GSE, and Genipin	Pretreatment	µTBS	Adper Single bond (3M ESPE, St. Paul, MN, USA) and One Step Plus	Distilled water at 37 °C for 24 h	Adhesive	Application of GA and GSE increased bond strength.
**Bedran-Russo, 2010** **[109]**	Human molars	EDC, MES, and bacterial collagenase from Clostridium histolyticum—type I	Pretreatment	µTBS	Adper Single bond (3M ESPE, St. Paul, MN, USA) and One Step Plus	Distilled water at 37 °C for 24 h or 12 months	Not measured	EDC preserved the bond strength after aging.
**Cova, 2011** **[116]**	Human molars	UVA-activated RF	Pretreatment	µTBS	XP Bond Adhesive (Dentsply)	Artificial saliva at 37 °C 24 h, 6 months, and 1 year	Adhesive	RF/UVA increased immediate and long-term bond strength.
**Kim, 2011** **[100]**	Human molars	EDTA	Pretreatment	µTBS	Adper Single Bond 2 (3M ESPE, St. Paul, MN, USA)	Distilled water at 37 °C for 24 h	NO	Dentin treatment with EDTA improved the Bond strength durability.
**Osorio, 2011** **[93]**	Human molars	Zinc	Incorporated into the adhesive system	µTBS	Single Bond Plus (3M ESPE, St Paul, MN, USA), and Clearfil SE Bond (Kuraray, Tokyo, Japan)	Distilled water at 37 °C for 24 h, 1 week, and 4 weeks.	Mixed	Zinc-doped resin did not affect bond strength.
**Islam, 2012** **[104]**	Human molars	HP and GSE	Incorporated into the adhesive system	µTBS	Clearfil SE bond (Kuraray Noritake Dental Inc. Tokyo, Japan)	Distilled water at 37 °C for 24 h	Cohesive/adhesive	Incorporation of HP had a positive influence on the immediate bond strength.
**Fang, 2017** **[79]**	Human molars	PRA	Pretreatment	µTBS	Adper Single Bond 2 (3M) and Prime&Bond NT (Dentsply)	Distilled water at 37 °C for 24 h	Mixed	PRA preconditioning improved resin–dentin bond strength.
**Fawzy, 2012** **[118]**	Human molars	RF	Pretreatment	µTBS	Adper Single Bond 2 (3M)	Distilled water at 37 °C for 24 h or 4 months	Non measured	RF maintained the bond strength after short-term water storage.
**Castellan, 2013** **[112]**	Human molars	GSE and cocoa seed	Pretreatment	µTBS	Single Bond Plus and One Step Plus	Simulated body fluid at 37 °C for 24 h, 3 months, 6 months, and 12 months.	Non measured	GSE enhanced immediate bond strength.
**Chiang, 2013** **[114]**	Human molars	RF/UVA, and GA	Pretreatment	µTBS	Scotch Bond Multipurpose (3M)	Distilled water to normal water at 37 °C for 24 h and thermocycling.	Adhesive / Mixed	RF/UVA treatment enhanced bond strength.
**Fawzy, 2013** **[117]**	Human molars and premolars	CH/RF	Pretreatment	µTBS	Adper Single Bond 2 (3M)	Distilled water at 37 °C for 24 h or 6 months.	Non measured	Significant improvement in bond strength was found.
**Islam, 2014** **[103]**	Human molars	HP	Incorporated into the adhesive system	µTBS	Clearfil SE bond (Kuraray Noritake Dental Inc. Tokyo, Japan	Distilled water at 37 °C for 24 h	Mixed/Cohesive	The incorporation HPN into the self-etching had a positive effect on the immediate bond strength.
**Liu, 2014** **[78]**	Human molars	PRA	Pretreatment	µTBS	Adper Single Bond 2 (3M ESPE, St. Paul, MN, USA)	Distilled water at 37 °C for 24 h	Mixed	PRA was effective stabilizing the adhesive–dentin bonding.
**Sabatini, 2014** **[91]**	Human molars	BAC	Incorporated into the adhesive system	µTBS	Adper Single Bond Plus (3M ESPE, St. Paul, MN, USA)	Artificial saliva at 37 °C for 24 h and 6 months	Adhesive + Mixed	BAC preserved the resin–dentin bond.
**Scheffel, 2015** **[90]**	Human molars	EDC	Pretreatment	µTBS	Single Bond 2 (3M ESPE, St. Paul, MN, USA)	Distilled water at 37 °C for 6 and 12 months	Adhesive + Mixed	EDC prevented resin–dentin bond degradation after aging.
**Singh, 2015** **[85]**	Human molars	EDTA and EDC	Pretreatment	SBS	G-Bond (GC Corp., Tokyo, Japan)	Distilled water at 37 °C for 24 h and 6 months	Mixed	EDC pretreatment resulted in preservation of resin–dentin bond strength.
**Yang, 2015** **[87]**	Human molars	EGCG	Pretreatment	µTBS	Adper Single Bond 2 (3M ESPE, St. Paul, MN, USA)	Thermocycling	Adhesive failure	Pretreatment with EGCG improve immediate bond strength and bond stability.
**Gerhardt, 2016** **[119]**	Human molars	EGCG and GTE	Pretreatment	µTBS	Clearfil SE Bond (Kuraray)	Distilled water at 37 °C for 24 h and 6 months	Cohesive in resin	Dentin pretreatment with EGCG andGTE increased bond strength.
**Hass, 2016** **[17]**	Human molars	PRA, RF, and GA	Pretreatment	µTBS	Single Bond Plus (SB)(3M ESPE, St. Paul, MN, USA), and Tetric N-Bond (TN) (Ivoclar Vivadent AG, Schaan, Liechtenstein)	Distilled water at 37 °C for 24 h and 18 months	Mixed	The PRA and GA treatments produced stable interfaces after aging.
**Joseph, 2016** **[102]**	Human molars	RF	Pretreatment	SBS	Adper Single Bond 2 (3M ESPE, St. Paul, MN, USA)	Distilled water at 37 °C for 24 h	NO	The pretreatment with RF improved the immediate bond strength.
**Leme-Kraus, 2017** **[76]**	Human molars	PRA	Pretreatment	µTBS	Experimental adhesives were applied	Simulated body fluid at 37 °C for 24 h and 1 year	NO	PRA produced a robust and stable adhesion.
**Loguercio, 2016** **[98]**	Human molars	MC	Pretreatment	µTBS	Adper Single Bond 2 [3M ESPE, St Paul, MN, USA), and Prime&Bond NT (Dentsply De Trey, Konstanz, Germany)	Distilled water at 37 °C for 24 h and 2 years	Adhesive/mixed fractures	MC retarded the degradation of resin dentin interfaces over a 24-month period.
**Neri, 2016** **[95]**	Human molars	EGCG	Pretreatment	µTBS	Adper Easy One (3M ESPE, St. Paul, MN, USA)	Distilled water at 37 °C for 24 h, 6 months, and 12 months	Mixed	Pretreatment with EGCG preserved the bond strength after aging.
**Venigalla, 2016** **[83]**	Human molars	RF, EDC, and PRA	Pretreatment	µTBS	Adper Single Bond Adhesive (3M ESPE)	Artificial saliva for 24 h at 37 °C and 6 months	Adhesive failure	Modification of dentin using collagen crosslinking improved the bond strength durability.
**Zhang, 2016** **[60]**	Human molars	EDC	Pretreatment	µTBS	Single Bond 2 (3M ESPE)	0.9% NaCl solution at 37 °C for 24 h and 90 days	Adhesive failure	The EDC-treated groups exhibited significantly greater bond strength.
**Zhang, 2016 (b)** **[82]**	Human molars	EDC	Pretreatment	µTBS	Single Bond 2, 3MESPE	0.9% NaCL at 37 °C for 24 h and 90 days	No	Treatment with EDC improved bond strength.
**Zhou, 2016** **[81]**	Human molars	GSE	Pretreatment	µTBS	Single Bond Plus (3M ESPE, St. Paul, MN, USA)	Distilled water at 37 °C for 24 h	No	GSE pretreatment allowed to use the dry bond technique without lowering the bond strength values.
**Abunawareg, 2017** **[106]**	Human molars	EDC and RF	Pretreatment	µTBS	Adper Single bond 2(3M ESPE, St. Paul, MN, USA)	Distilled water at 37 °C for 24 h, 6 months, and 12 months	Mixed	Collagen crosslinking induced by EDC and RF improved the durability of the resin–dentin bond.
**Feiz, 2017** **[80]**	Human molars	GSE	Pretreatment	SBS	Adper Single Bond 2 (3M)	Distilled water at 37 °C for 24 h	Non measured	GSE material did not improve the bond strength.
**Lee, 2017** **[99]**	Human molars	GA	Incorporated into the adhesive system	µTBS	Gluma Comfort Bond and Desensitizer and iBond Total Etch (both from Heraeus Kulzer)]	Distilled water at 37 °C for 24 h	Mixed	Collagen crosslinking with GA stabilized the bond strength after aging.
**Paulose, 2017** **[92]**	Human molars	GSE	Pretreatment	µTBS	AdperTM Single Bond 2 (3M ESPE, St. Paul, MN, USA), and Single Bond Universal Adhesive (3M ESPE, St. Paul, MN, USA)	Distilled water at 37 °C for 24 h and 1 year	No	GSE was effective only by using it in the dry bond technique.
**Yu, 2017** **[84]**	Human molars	EGCG	Incorporated into the adhesive system	µTBS	Single Bond 2 (3M ESPE)	Deionized water at 37 °C for 24 h and thermocycling.	No	EGCG-containing adhesive improved the long-term bond strength.
**Bharti, 2018** **[110]**	Human molars	SA and PRA	Pretreatment	SBS	One Coat Bond SL (Coltene)	Distilled water to normal water at 37 °C for 24 h and thermocycling.	Non measured	Treatment with collagen crosslinking agent increased the bond strength-
**Mazzoni, 2018** **[97]**	Human molars	EDC	Pretreatment	µTBS	Clearfil SE Bond (Kuraray Dental, Osaka, Japan), and XP Bond (Dentsply DeTrey GmbH, Konstanz, Deustche; abbreviation: XPB)	Artificial saliva at 37 °C for 24 h and 1 year.	Mixed	The use of EDC improved the bond strength over the time.
**Sun, 2018** **[86]**	Human molars	EGCG	Pretreatment	µTBS	Adper Single Bond 2 (3M ESPE, St. Paul, MN, USA)	Thermocycling	No	EGCG preconditioning enhanced resin–dentin bond durability.
**Chaharom, 2019** **[113]**	Human molars	GR	Non specified	µTBS	All-Bond 3, One-Step Plus, Clearfil SE Bond (Kuraray) and Clearfil S3 (Kuraray)	Thermocycling	Adhesive / Mixed	GR had no effect on the immediate bond strength.
**Kasraei, 2019** **[101]**	Human premolars	RF	Pretreatment	µTBS	Clearfil SE Bond adhesive (Kuraray, Tokyo, Japan)	Distilled water at 37 °C for 24 h and thermocycling.	Mixed	Treatment of dentin with RF had a negative impact on the bond strength.
**Silva, 2019** **[89]**	Human molars and premolars	GSE	Pretreatment	SBS	ClearfilTM SE Bond (Kuraray, Medical Inc., Tokyo, Japan)	Thermocycling	No	The application of GSE did not improve the bond strength.
**Tang, 2016** **[88]**	Human molars	EDC	Pretreatment	SBS	Single Bond 2 (3M ESPE, St. Paul, MN, USA)	Distilled water at 37 °C for 24 h	Adhesive failure	EDC favored the bond strength.
**Venkatachalam, 2019** **[121]**	Human premolars	RF	Pretreatment	SBS	Adper Single Bond 2 (3M ESPE, St. Paul, MN, USA)	Distilled water at 37 °C and thermocycling.	No	Pretreatment by application of RF improved the shear bond strength.
**Baena, 2020** **[107]**	Human molars	CH	Pretreatment	µTBS	Optibond FL (Kerr)	Artificial saliva at 37 °C for 24 h or thermocycling.	Adhesive	Application of CH did not improve the long-term bond strength.
**Comba, 2019** **[115]**	Human molars	EDC	Pretreatment	µTBS	Single Bond Universal (3M)	Artificial saliva at 37 °C 24 h and 1 year	Adhesive	The use of EDC preserved the stability of the adhesive interface.
**Fu, 2020** **[124]**	Human molars	Modified RF	Incorporated into the adhesive system	µTBS	Single Bond Universal (3M) and Zipbond	Distilled water at 37 °C for 1 week and 6 months	Mixed	RF modified adhesives improved the long-term bond strength.
**Li, 2020** **[125]**	Human molars	DMA	Pretreatment	µTBS	Adper Single Bond 2 (3M ESPE, St. Paul, MN, USA)	Distilled water at 37 °C for 24 h and thermocycling	Mixed	Pretreatment with DMA preserved the bond strength.
**Nivedita, 2019** **[123]**	Human premolars	CH and PRA	Pretreatment	SBS	Adper Single Bond-2 (3M ESPE, St. Paul, MN, USA)	Distilled water at 37 °C for 24 h and thermocycling.	NO	Application of Chitosan and PRA improved the bond strength.
**Baldion, 2021** **[108]**	Human molars	EDTA, GA, PRA, and MY.	Pretreatment	µTBS	Adper Single bond 2 (3M ESPE, St. Paul, MN, USA)	Artificial saliva at 37 °C for 24 h and 18 months.	Adhesive	Use of GA and PRA preserved the bond strength after aging.
**Bourgi, 2021** **[111]**	Human molars	RB	Incorporated into the adhesive system	µTBS	Prime&Bond Active (Dentsply)	Distilled water at 37 °C for 24 h	Non measured	RB helped in maintaining bond strength.
**Freitas, 2021** **[77]**	Human molars	DX	Incorporated into the adhesive system	µTBS	Adper Single Bond 2 (3M)	Distilled water at 37 °C for 24 h and 1 year.		DX adhesives maintained the bond strength after aging.

HEMA: hydroxyethyl methacrylate; GA: glutaraldehyde; EDTA: ethylenediaminetetraacetic acid; GSE: grape seed extract; EDC: 1-ethyl-3-(3-dimethylaminopropyl)carbodiimide hydrochloride; MES: N-hydroxysuccinimide-NHS, 2-(N-morpholino)ethanesulfonic acid; RF: riboflavin; HP: hesperidin; PRA: proanthocyanidin; RF/UVA: riboflavin-ultraviolet; CH/RF: chitosan/riboflavin; BAC: benzalkonium chloride; EDC: carbodiimide; EGCG: epigallocatechin gallate; GTE: green tea extract; MC: minocycline; SA: sodium ascorbate; DMA: dopamine methacrylamide; MY: myricetin; RB: ribose; DX: doxycline; GR: galardin; TBS: tensile bond strength; µTBS: micro-tensile bond strength; SBS: shear bond strength.

**Table 4 cells-11-02417-t004:** Risk of bias of the included studies.

Study	Specimen Randomization	Single Operator	Operator Blinded	Control Group	Standardized Specimens	Failure Mode	Manufacturer’s Instructions	Sample size Calculation	Risk of Bias
Abunawareg, 2017 [106]	YES	NO	NO	YES	YES	YES	YES	NO	Medium
Al-Ammar, 2009 [105]	YES	NO	NO	YES	YES	YES	YES	NO	Medium
Baena, 2020 [107]	NO	NO	NO	YES	YES	YES	YES	NO	Medium
Baldion, 2021 [108]	YES	NO	NO	YES	YES	YES	YES	NO	Medium
Bedran-Russo 2010 [109]	YES	NO	NO	YES	YES	NO	YES	NO	Medium
Bharti, 2018 [110]	YES	NO	NO	YES	YES	NO	YES	NO	Medium
Bourgi, 2021 [111]	YES	NO	NO	YES	YES	NO	YES	YES	Medium
Castellan, 2013 [112]	NO	NO	NO	YES	YES	NO	YES	NO	High
Chaharom, 2019 [113]	YES	NO	NO	YES	YES	YES	YES	NO	Medium
Chiang, 2013 [114]	YES	NO	NO	YES	YES	YES	YES	NO	Medium
Comba, 2019 [115]	YES	YES	NO	YES	YES	YES	YES	NO	Medium
Cova 2011 [116]	YES	NO	NO	YES	YES	YES	YES	NO	Medium
Fu, 2020 [124]	YES	NO	YES	YES	YES	YES	YES	NO	Medium
Fang, 2017 [79]	YES	NO	NO	YES	YES	YES	YES	NO	Medium
Fawzy, 2012 [118]	NO	NO	NO	YES	YES	YES	YES	NO	Medium
Fawzy, 2013 [117]	YES	NO	NO	YES	YES	NO	YES	NO	Medium
Feiz, 2017 [80]	NO	NO	NO	YES	YES	NO	YES	NO	High
Freitas, 2021 [77]	YES	NO	NO	YES	YES	NO	YES	NO	Medium
Gerhardt 2016 [119]	YES	NO	NO	YES	YES	YES	YES	NO	Medium
Hass, 2016 [17]	YES	NO	YES	YES	YES	YES	YES	NO	Medium
Islam, 2012 [104]	NO	NO	NO	YES	YES	YES	YES	NO	Medium
Islam, 2014 [103]	NO	NO	NO	YES	YES	YES	YES	NO	Medium
Joseph, 2016 [102]	YES	NO	NO	YES	YES	NO	YES	NO	Medium
Kasraei, 2019 [101]	YES	YES	NO	YES	YES	YES	YES	NO	Medium
Kim, 2011 [100]	YES	NO	NO	YES	YES	NO	YES	NO	Medium
Lee, 2017 [99]	YES	YES	NO	YES	YES	YES	YES	NO	Medium
Leme-Kraus, 2017 [76]	NO	NO	NO	YES	YES	NO	YES	NO	High
Li, 2020 [125]	YES	NO	NO	YES	YES	YES	YES	NO	Medium
Liu, 2014 [78]	YES	NO	NO	YES	YES	YES	YES	NO	Medium
Loguercio, 2016 [98]	YES	YES	NO	YES	YES	YES	YES	NO	Medium
Mazzoni, 2018 [97]	YES	YES	NO	YES	YES	YES	YES	NO	Medium
Munksgaard, 2002 [96]	NO	NO	NO	YES	YES	NO	YES	NO	High
Neri, 2016 [95]	YES	NO	NO	YES	YES	YES	YES	NO	Medium
Nivedita, 2019 [123]	NO	NO	NO	YES	YES	NO	YES	NO	High
Osorio, 2005 [94]	NO	NO	NO	YES	YES	YES	YES	NO	Medium
Osorio, 2011 [93]	NO	NO	NO	YES	YES	YES	YES	NO	Medium
Paulose, 2017 [92]	YES	NO	NO	YES	YES	NO	YES	NO	Medium
Sabatini, 2014 [91]	YES	NO	NO	YES	YES	YES	YES	NO	Medium
Scheffel, 2015 [90]	YES	NO	NO	YES	YES	YES	YES	NO	Medium
Silva, 2019 [89]	YES	NO	NO	YES	YES	NO	YES	NO	Medium
Singh, 2015 [85]	YES	NO	NO	YES	YES	YES	YES	NO	Medium
Sun, 2018 [86]	NO	NO	NO	YES	YES	NO	YES	NO	High
Tang, 2016 [88]	YES	NO	NO	YES	YES	YES	YES	NO	Medium
Venigalla, 2016 [83]	YES	NO	NO	YES	YES	YES	YES	NO	Medium
Venkatachalam, 2019 [121]	YES	NO	NO	YES	YES	NO	YES	NO	Medium
Yang, 2015 [87]	YES	NO	NO	YES	YES	YES	YES	NO	Medium
Yu, 2017 [84]	YES	NO	NO	YES	YES	NO	YES	YES	Medium
Zhang, 2016 [60]	NO	NO	NO	YES	YES	YES	YES	NO	Medium
Zhang, 2016 (b) [82]	NO	NO	NO	YES	YES	NO	YES	NO	High
Zhou, 2016 [81]	NO	NO	NO	YES	YES	NO	YES	NO	High

## Data Availability

The data that support the findings of this study are available from the first author (L.H.) upon reasonable request.

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
