# Peer review of "Effect of Collagen Crosslinkers on Dentin Bond Strength of Adhesive Systems: A Systematic Review and Meta-Analysis"

_cells, 2022, doi:10.3390/cells11152417_

Round 1

Reviewer 1 Report

Through a systematic review and meta-analysis, the manuscript demonstrates the role of crosslinking agents in resin-dentin bond strength when used as adhesives or pretreatments on dentin surfaces. It is an excellent paper that can increase understanding by comparing the bond strength of various adhesive formulation products. However, there are a few suggestions to improve it as follows.;

 Comment;

1.  Please increase the figure resolution.

2.  Table 2 seems easier to understand if rearranged by year of study. In addition, please display the contents of Table 2 at the top of the next page.

Author Response

Dear Sir or Madam,

Thank you for your review. We are very thankful for all your comments that enhanced the quality of manuscript. All your suggestions were addressed accordingly and marked in text.

Through a systematic review and meta-analysis, the manuscript demonstrates the role of crosslinking agents in resin-dentin bond strength when used as adhesives or pretreatments on dentin surfaces. It is an excellent paper that can increase understanding by comparing the bond strength of various adhesive formulation products. However, there are a few suggestions to improve it as follows.;

 Comment;

  1. Please increase the figure resolution.

 R: Thank you for the comment. The actual resolution of the images is 300 pixels/inch, that it’s enough for online visualization. For this version of the manuscript, we increased the size of the images for improving their visualization in the PDF file. Thank you!

  1. Table 2 seems easier to understand if rearranged by year of study. In addition, please display the contents of Table 2 at the top of the next page.

R: Thank you for your comment. The table was now corrected according to your recommendation.

Reviewer 2 Report

The article is nicely written and informative.

I suggest some minor improvements:

lines 63 to 65 - please reformulate the sentence as it is not clear what the authors intend to say.

line 75 - researchers 

line 307 - lessen

line 394 - counteract

lines 394 to 401 - I would remove this paragraph and pace it at the end of the discussion section 

Author Response

Dear Sir or Madam,

Thank you for your review. We are very thankful for all your comments that enhanced the quality of manuscript. All your suggestions were addressed accordingly and marked in text.

The article is nicely written and informative.

R: Thank you so much for your comment.

I suggest some minor improvements:

  1. lines 63 to 65 - please reformulate the sentence as it is not clear what the authors intend to say.

R: Thank you for the comment. The sentence was now reformulated.

  1. line 75 – researchers; line 307 – lessen; line 394 – counteract

R: Thank you, these mistakes were now corrected.

  1. lines 394 to 401 - I would remove this paragraph and pace it at the end of the discussion section 

R: Thank you for the comment. The paragraph was now moved according to your recommendation.

Reviewer 3 Report

Comments:

1. The abstract could be more comprehensive by focusing on important parts. Please rewrite the abstract and try to emphasize the significant parts.

2. Keywords should be determined by the appropriate MeSH Terms in NCBI.

3. Presenting Inclusion and Exclusion criteria for selecting the articles in Table format can be helpful.

4. Due to the in vitro nature of evaluated articles for systematic review, the risk of bias assessment should be evaluated by modified tools such as mentioned in previous articles:

a)AlShwaimi, E., Bogari, D., Ajaj, R., Al‐Shahrani, S., Almas, K., & Majeed, A. (2016). In vitro antimicrobial effectiveness of root canal sealers against Enterococcus faecalis: A systematic review. Journal of Endodontics, 42(11), 1588–1597.

b) Samiei, M., Shirazi, S., Azar, F. P., Fathifar, Z., Ghojazadeh, M., & Alipour, M. (2019). The effect of different mixing methods on the properties of calcium‐enriched mixture cement: A systematic review of in vitro studies. Iranian Endodontic Journal, 14(4), 240–246. 

c)Neelakantan, P., Ahmed, H., Wong, M., Matinlinna, J., & Cheung, G. (2018). Effect of root canal irrigation protocols on the dislocation resistance of mineral trioxide aggregate‐based materials: A systematic review of laboratory studies. International Endodontic  Journal, 51(8), 847–861.

If you have already used these tools, please mention it in your article and cite the above articles.

5. In table 2, it is suggested that authors use abbreviations, and those abbreviations are going to be explained in the table caption. It could be helpful for the uniform structure of the table.

6. In Table 2, It is suggested that the type of tooth mentioned in the uniform pattern, for example, extracted human posterior teeth is not clear; please define all samples as molar, premolar,… ,and mention the source of the tooth. Moreover, due to the in vitro nature of evaluated studies, it is clear that all teeth should be extracted. I wonder if it might not be necessary to mention it.

7. In Table 2, The Type of teeth is not mentioned in Gerhardet 2016 study, while it was mentioned in the study: Thirty-two human third molars, extracted for reasons not related to those of the present study 

Please kindly note that any kind of missing data in systematic reviews for any reason might doubt the accuracy of the search and data extraction procedure. Please before submitting check all extracted data and revise them carefully.

8. There are several similar articles that can improve your discussion; please try to improve your discussion with those articles. Furthermore, there is a systematic review in 2022 with with different results (Silva JC, Cetira Filho EL, de Barros Silva PG, Costa FW, Saboia VD. Is dentin biomodification with collagen cross-linking agents effective for improving dentin adhesion? A systematic review and meta-analysis. Restorative dentistry & endodontics. 2022 May;47(2).) Please compare it with yours and discuss its differences and results in discussion part.

9.  Failure mode is reported both as a risk of bias and extracted data in table 2. It might be better if the failure mode ommited from risk of bias table.

10. It is suggested that the outcome of each study is briefly expressed in table 2 under the main result section rather than writing primary, and secondary outcomes. Please check the similar study:

Neelakantan, P., Ahmed, H., Wong, M., Matinlinna, J., & Cheung, G. (2018). Effect of root canal irrigation protocols on the dislocation resistance of mineral trioxide aggregate‐based materials: A systematic review of laboratory studies. International Endodontic 

11. The writing of the manuscript could be improved by an expert English writer.

Author Response

Dear Sir or Madam,

Thank you for your review. We are very thankful for all your comments that enhanced the quality of manuscript. All your suggestions were addressed accordingly and marked in text.

Comments:

  1. The abstract could be more comprehensive by focusing on important parts. Please rewrite the abstract and try to emphasize the significant parts.

 R: Thank you for the comment. The abstract was rewritten and the results section now focus in the significant parts.

  1. Keywords should be determined by the appropriate MeSH Terms in NCBI.

 R: Thank you for the comment. Keywords were updated.

  1. Presenting Inclusion and Exclusion criteria for selecting the articles in Table format can be helpful.

 R: Thank you for the suggestion. The Table 2 was added explaining the inclusion and exclusion criteria used.

  1. Due to the in vitro nature of evaluated articles for systematic review, the risk of bias assessment should be evaluated by modified tools such as mentioned in previous articles:

a)AlShwaimi, E., Bogari, D., Ajaj, R., Al‐Shahrani, S., Almas, K., & Majeed, A. (2016). In vitro antimicrobial effectiveness of root canal sealers against Enterococcus faecalis: A systematic review. Journal of Endodontics, 42(11), 1588–1597.

b) Samiei, M., Shirazi, S., Azar, F. P., Fathifar, Z., Ghojazadeh, M., & Alipour, M. (2019). The effect of different mixing methods on the properties of calcium‐enriched mixture cement: A systematic review of in vitro studies. Iranian Endodontic Journal, 14(4), 240–246. 

c)Neelakantan, P., Ahmed, H., Wong, M., Matinlinna, J., & Cheung, G. (2018). Effect of root canal irrigation protocols on the dislocation resistance of mineral trioxide aggregate‐based materials: A systematic review of laboratory studies. International Endodontic  Journal, 51(8), 847–861.

If you have already used these tools, please mention it in your article and cite the above articles.

R: Actually, we used a modified tool for the risk of bias assessment, and this was explained in the material and methods section (line 154). The modified tool used here was already used in other works of the teamwork and it’s properly cited in the manuscript. Thank you!

5. In table 2, it is suggested that authors use abbreviations, and those abbreviations are going to be explained in the table caption. It could be helpful for the uniform structure of the table.

R: Thank you for the suggestion. We modified the Table according to your suggestion.

  1. In Table 2, It is suggested that the type of tooth mentioned in the uniform pattern, for example, extracted human posterior teeth is not clear; please define all samples as molar, premolar,… ,and mention the source of the tooth. Moreover, due to the in vitro nature of evaluated studies, it is clear that all teeth should be extracted. I wonder if it might not be necessary to mention it.

R: Thank you for the comment. Information from Table 2 was checked and updated.

  1. In Table 2, The Type of teeth is not mentioned in Gerhardet 2016 study, while it was mentioned in the study: Thirty-two human third molars, extracted for reasons not related to those of the present study 

Please kindly note that any kind of missing data in systematic reviews for any reason might doubt the accuracy of the search and data extraction procedure. Please before submitting check all extracted data and revise them carefully.

R: Thank you for the comment. Information from Table 2 was checked and updated.

  1. There are several similar articles that can improve your discussion; please try to improve your discussion with those articles. Furthermore, there is a systematic review in 2022 with with different results (Silva JC, Cetira Filho EL, de Barros Silva PG, Costa FW, Saboia VD. Is dentin biomodification with collagen cross-linking agents effective for improving dentin adhesion? A systematic review and meta-analysis. Restorative dentistry & endodontics. 2022 May;47(2).) Please compare it with yours and discuss its differences and results in discussion part.

R: Thank you for the comment. The discussion section was now improved, and novel studies were now added.

9. Failure mode is reported both as a risk of bias and extracted data in table 2. It might be better if the failure mode ommited from risk of bias table

R: As you kindly mention, the failure mode is reported in the both characteristics of the studies included, and in the Risk of Bias analysis. However, they are reporting different things. While in the Table 2 is described which type of failure mode was predominant in the specimens tested in such study, the Risk of Bias table express if the authors performed or no such analysis.

According to the Academy of Dental Materials guidance on in vitro testing of dental composite bonding effectiveness to dentin/enamel using micro-tensile bond strength (TBS) approach (DOI: http://dx.doi.org/10.1016/j.dental.2016.11.015), the failure analysis is recommended for the processing of the tested specimens. In this sense, we believe that including this parameter within the risk of bias table is mandatory for the evaluation of the quality of each manuscript.

  1. It is suggested that the outcome of each study is briefly expressed in table 2 under the main result section rather than writing primary, and secondary outcomes. Please check the similar study:

Neelakantan, P., Ahmed, H., Wong, M., Matinlinna, J., & Cheung, G. (2018). Effect of root canal irrigation protocols on the dislocation resistance of mineral trioxide aggregate‐based materials: A systematic review of laboratory studies. International Endodontic 

R: Thank you for the comment. The Table was modified according to your suggestion.

11. The writing of the manuscript could be improved by an expert English writer.

R: English was revised by a proofreading English company. Thank you!

Round 2

Reviewer 3 Report

It is acceptable after revision.